# DECO: Unleashing the Potential of ConvNets for Query-based Detection and Segmentation

**Xinghao Chen**[1]*, **Siwei Li**[1,2]*, **Yijing Yang**[1], **Yunhe Wang**[1]

[1] Huawei Noah's Ark Lab    [2] Tsinghua University
{xinghao.chen, yunhe.wang}@huawei.com

## Abstract

Transformer and its variants have shown great potential for various vision tasks in recent years, including image classification, object detection and segmentation. Meanwhile, recent studies also reveal that with proper architecture design, convolutional networks (ConvNets) also achieve competitive performance with transformers. However, no prior methods have explored to utilize pure convolution to build a Transformer-style Decoder module, which is essential for Encoder-Decoder architecture like Detection Transformer (DETR). To this end, in this paper we explore whether we could build query-based detection and segmentation framework with ConvNets instead of sophisticated transformer architecture. We propose a novel mechanism dubbed InterConv to perform interaction between object queries and image features via convolutional layers. Equipped with the proposed InterConv, we build Detection ConvNet (DECO), which is composed of a backbone and convolutional encoder-decoder architecture. We compare the proposed DECO against prior detectors on the challenging COCO benchmark. Despite its simplicity, our DECO achieves competitive performance in terms of detection accuracy and running speed. Specifically, with the ResNet-18 and ResNet-50 backbone, our DECO achieves 40.5% and 47.8% AP with 66 and 34 FPS, respectively. The proposed method is also evaluated on the segment anything task, demonstrating similar performance and higher efficiency. We hope the proposed method brings another perspective for designing architectures for vision tasks. Codes are available at `https://github.com/xinghaochen/DECO` and `https://github.com/mindspore-lab/models/tree/master/research/huawei-noah/DECO`.

## 1 Introduction

Object detection and segmentation are among the most foundational computer vision tasks and are essential for many real-world applications (Ren et al., 2015; Redmon & Farhadi, 2017; 2018; Bochkovskiy et al., 2020). The object detection pipeline has been developed rapidly, especially in the era of deep learning. Faster R-CNN (Ren et al., 2015) is one of the most typical two-stage object detectors, which utilizes a coarse-to-fine framework for bounding box prediction. Meanwhile, one-stage detectors like SSD (Liu et al., 2016), YOLO series (Redmon & Farhadi, 2017; 2018; Bochkovskiy et al., 2020) or FCOS (Tian et al., 2019) simplify the detection pipeline by directly predicting the objects of interest from the image features. Most of the above object detectors are built upon convolutional neural networks (ConvNets) and typically the Non-maximum Suppression (NMS) strategy is utilized for post-processing to remove duplicated detection results.

The advancement of deep neural architectures have been benefiting the task of object detection. For example, more powerful architectures usually bring considerably significant improvement for the detection performance (Li et al., 2018; Liu et al., 2020; Gao et al., 2019; Guo et al., 2020). Recently the emergence of vision transformer and its variants (Dosovitskiy et al., 2021; Liu et al., 2021; Touvron et al., 2021; Wang et al., 2021b) have shown prominent performances on image classification tasks and have built a solid foundation for the object detection field. Carion *et al*. (Carion et al.,

---

* Equal contribution. Correspondence to X. Chen and Y. Wang.

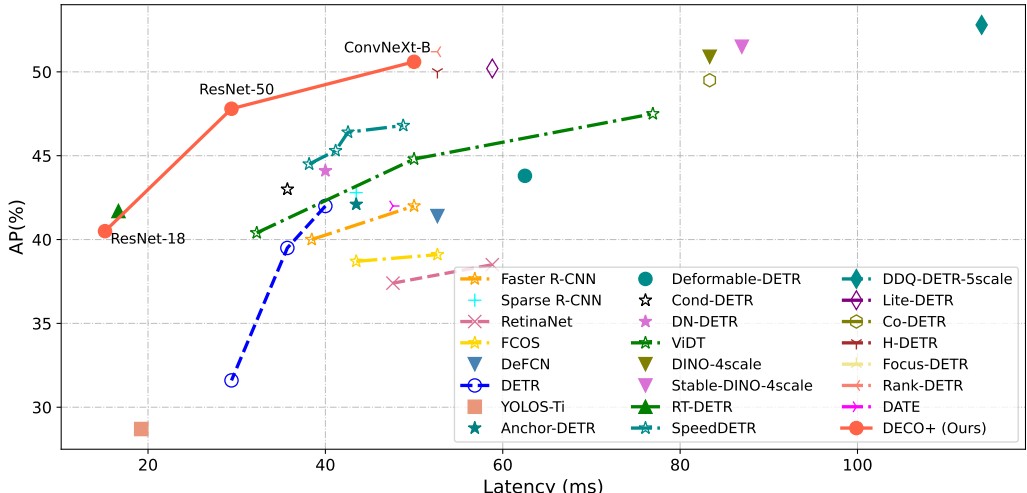

Figure 1: Comparisons of our proposed Detection ConvNets (DECO) and recent detectors on COCO *val* set. The latency is measured on a NVIDIA V100 GPU.

2020) proposes the Detection Transformer (DETR) that refactors the object detection pipeline as a set prediction problem and directly obtains a fixed set of objects via a transformer encoder-decoder architecture. This design enables DETR to get rid of the complicated NMS post-processing module and results in a query-based end-to-end object detection pipeline. There are quite a lot of variants to improve DETR via different aspects, *e.g.*, training convergence (Meng et al., 2021; Gao et al., 2021), multi-scale features and deformable attention (Zhu et al., 2021) or better query strategy (Li et al., 2022; Liu et al., 2022a; Wang et al., 2022b; Zhang et al., 2022).

Despite the strong performance of transformer, it does introduce more challenges for AI chips (Tu et al., 2022). More specifically, attention layers introduce dynamic memory access whose weights and inputs are both generated during runtime, while convolutional layers have static memory access. Therefore, it is still common that certain operators like attention module is not well supported in some AI chips, which is a big challenge in industry.

Meanwhile, some recent work rethinks the strong performance and reveal that the pure ConvNets could also achieve competitive performance via proper architecture design (Liu et al., 2022b; Yu et al., 2022). For example, ConvNeXt (Liu et al., 2022b) competes favorably with vision transformers like Swin Transformer (Liu et al., 2021) in terms of accuracy and computational cost. However, these methods mainly focus on encoder part of transformer, in which self-attention is utilized and could be replaced by convolution with careful design. These motivate us to explore one important question in this paper: *could we obtain an architecture via pure ConvNets but still enjoys the excellent properties similar to attention?*

In this paper, we propose a novel mechanism dubbed **InterConv** to perform interaction between object queries and image features via pure convolutional layers. We abstract the general architecture of a decoder and divide it into two components, *i.e.*, Self-Interaction Module (SIM) and Cross-Interaction Module (CIM). In transformer-based models, the SIM and CIM are implemented with multi-head self-attention and cross-attention, while they are built upon our proposed InterConv in our method. The Self-InterConv is stacked with simple depthwise and $1 \times 1$ convolutions. A novel Cross-InterConv mechanism is further carefully designed to perform interaction between object queries and image features via convolutional layers as well as simple upsampling and pooling operations.

Equipped with the proposed InterConv, we develop **De**tection **Co**nvNet (**DECO**), which is a simple yet effective query-based end-to-end object detection framework. Our DECO model enjoys the similar favorable attributes as DETR. For example, using the mechanism of object query, our DECO directly obtains a fixed set of object predictions and also discards the NMS procedure. Moreover, it is stacked with only standard convolutional layers and does not rely on any sophisticated attention modules. To achieve this goal, we first carefully revisit the design of DETR and propose the DECO encoder and decoder architectures as shown in Fig. 3. The DECO encoder is built upon ConNeXt blocks and no positional encodings are necessary since ConvNets are variant to input permutation, while the DECO decoder is built up with InterConv.

We evaluate the proposed DECO on the challenging object detection benchmark, *i.e.*, COCO (Lin et al., 2014). Experimental results demonstrate that our DECO achieves competitive performance in terms of detection accuracy and running speed, as shown in Fig. 1. Specifically, with the ResNet-18 and ResNet-50 backbone (He et al., 2016), our DECO achieves $40.5\%$ and $47.8\%$ AP with 66 and 34 FPS, respectively and outperforms the DETR model. Extensive ablation studies are also conducted to provide more discussions and insights about the design choices. We also apply the proposed method into the popular segment anything task. Our DECO-TinySAM obtains quite similar performance and higher efficiency on mobile phone with the TinySAM baseline, demonstrating the effectiveness of our proposed method.

## 2 RELATED WORK

**Object Detection.** Object detection is one of the most foundational computer vision task and has attracted large amount of research interest from the computer vision community. The object detection pipeline has been developed rapidly, especially in the era of deep learning. Faster R-CNN (Ren et al., 2015) is one of the most typical two-stage object detectors, which first generates region proposal and extracts regional features for final bounding box prediction. Two-stage detection pipeline has been improve from various aspects (Pang et al., 2019; Cai & Vasconcelos, 2018). Meanwhile, one-stage detectors like SSD (Liu et al., 2016), YOLO series (Redmon et al., 2016; Redmon & Farhadi, 2017; 2018; Bochkovskiy et al., 2020), CenterNet (Zhou et al., 2019; Duan et al., 2019) or FCOS (Tian et al., 2019) simplify the detection pipeline by directly predicting the objects of interest from the image features (Li et al., 2019; Lu et al., 2019; Zhu et al., 2019; Kong et al., 2019; Zhu et al., 2020; Law et al., 2020; Zhang et al., 2020).

**Transformer-based End-to-End Detectors.** The pioneering work DETR (Carion et al., 2020) utilizes a transformer encoder-decoder architecture and models the object detection as a set prediction problem. It directly predicts a fixed number of objects and get rid of the need for hand-designed non-maximum suppression (NMS) (Neubeck & Van Gool, 2006). More follow-up studies (Meng et al., 2021; Gao et al., 2021; Dai et al., 2021a; Wang et al., 2022b) have made various optimizations and extensions based on the original DETR and achieve strong detection performance. For example, Deformable DETR (Zhu et al., 2021) only attends to a small set of key sampling points by introducing multi-scale deformable self/cross-attention to improve the detection accuracy as well as the training convergence. DAB-DETR (Liu et al., 2022a) improves DETR by using box coordinates as queries in decoder. DN-DETR and DINO (Li et al., 2022; Zhang et al., 2022) introduce several novel techniques, including query denoising, mixed query selection *etc*., to achieve strong detection performance. RT-DETR (Lv et al., 2023) designs the first real-time end-to-end detector, in which an efficient multi-scale hybrid encoder and an IoU-aware query selection are proposed. One of the most important properties for DETR-based detectors is the query-based scheme for producing the final predictions, which streamlines the detection pipeline and make it an end-to-end detector.

**ConvNet-based End-to-End Detectors.** Inspired by the success of transformer-based detector like DETR variants, several studies also attempt to remove the post-processing NMS by introducing one-to-one assignment strategy (Sun et al., 2021a; Wang et al., 2021a) and set prediction loss (Sun et al., 2021b). OneNet (Sun et al., 2021a) systemically explores the importance of classification cost in one-to-one matching and applies it on typical ConvNet-based detectors like RetinaNet (Lin et al., 2017) and FCOS (Tian et al., 2019). DeFCN (Wang et al., 2021a) introduces a new strategy of label assignment to enhance the matching cost. Sparse R-CNN (Sun et al., 2021b) integrates the fixed number of learnable anchor to a two-stage detection pipeline. However, it interacts query and RoI feature by the dynamic head which is a kind of learnable matrix multiplication.

ConvNets have been demonstrated to have competitive performance on various tasks and are deployment-friendly in most hardware platforms (Liu et al., 2022b; Yu et al., 2022). Cheng *et al*.proposed SparseInst (Cheng et al., 2022), an efficient and fully convolutional framework for real-time instance segmentation. SparseInst utilizes a sparse set of instance activation maps to predict objects in end-to-end style, which could be viewed as an alternative for the query-based mechanism of DETR. In this paper we would like to design a DETR-like detection pipeline but built with standard convolutions, which could inherit both the advantages of ConvNets and the favorable properties of the DETR framework. Compared with SparseInst, our method follows the similar query-base mechanism of DETR, and also demonstrates good generalization capability like segment anything models.

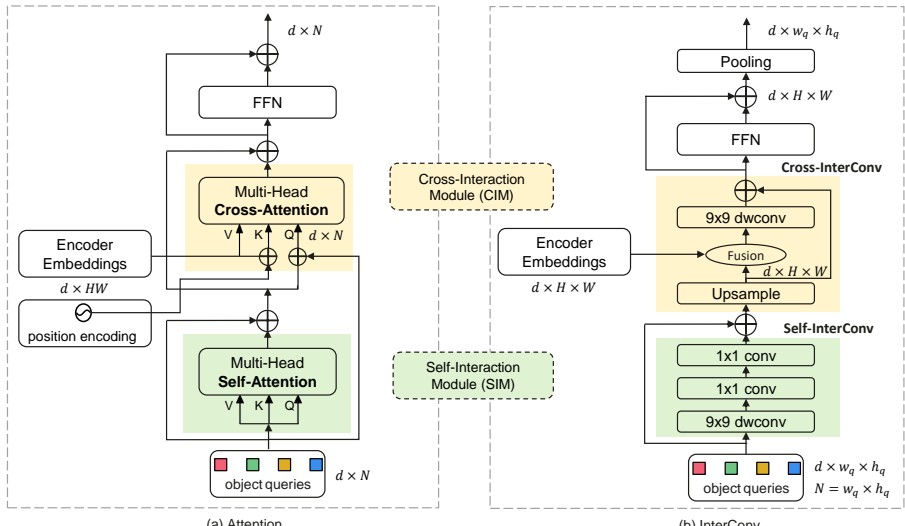

Figure 2: The attention-based decoder and our proposed InterConv. We abstract the general architecture of decoder and divide it into two components, *i.e.*, Self-Interaction Module (SIM) and Cross-Interaction Module (CIM). In DETR, the SIM and CIM are implemented with multi-head self-attention and cross-attention mechanism, while in our proposed DECO, the SIM is stacked with simple depthwise and $1 \times 1$ convolutions. We further propose a novel CIM mechanism for our DECO to perform interaction between object queries and image features via convolutional layers as well as simple upsampling and pooling operations.

## 3 APPROACH

In this section, we first introduce the design of our proposed InterConv, which works similarly with attention mechanism but is simply built with pure convolution. We then provide details about utilizing InterConv to develop efficient models for detection and segmentation.

### 3.1 INTERCONV: BUILDING ATTENTION-LIKE MECHANISM WITH CONVOLUTION

Given a small set of object queries, the decoder in transformer-based models like DETR or SAM aims to reason the relations of the objects and the global image feature. As shown in Fig. 2 (a), each layer in transformer decoder is mainly composed of a self-interaction module (SIM) and a cross-interaction module (CIM). The SIM in original DETR is a multi-head self-attention layer and is responsible for interacting information between the object queries. The CIM is the essential part for DETR decoder, which consists of cross-attention layers to perform interaction between the image embeddings from the output of encoder and the object queries. In this way, the object queries can attend to the global image feature and capture the essential information for each predicted objects. In this section, we aim to explore how to build an attention-like mechanism with ConvNets while maintaining the capability similar to attention.

**Self-InterConv for SIM**. Take DETR as an example, given $N$ object queries $o \in \Re^{N \times d}$, we first reshape the queries to $\Re^{w_q \times h_q \times d}$ and feed them into convolutional layers. For example, if we have $N = 100$ object queries, the query embeddings are reshaped into $\Re^{10 \times 10 \times d}$. More design choices of reshaping will be discussed in ablation studies. As shown in Fig. 2 (b), the SIM part for DECO decoder is quite similar to the design scheme of DECO encoder, where stacking the depthwise convolution and $1 \times 1$ convolution could lead to strong capability similar to the self-attention mechanism. We utilize a large kernel convolution up to $9 \times 9$ to perform long-range perceptual feature extraction.

**Cross-InterConv for CIM**. The CIM mainly takes two features as input, *e.g.*, the image feature embeddings from the output of encoder ($z_e \in \Re^{d \times H \times W}$), and the object query embeddings produced from the SIM part ($o \in \Re^{w_q \times h_q \times d}$). The cross-attention mechanism in DETR decoder allows each object query to interact with the image features to capture necessary information for object prediction. However, using ConvNets to perform such kind of interaction is not so intuitive. As shown in

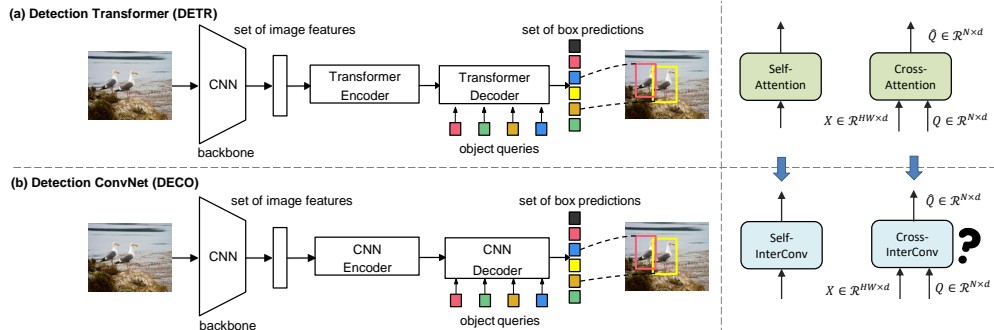

Figure 3: The overall architecture of DETR (Carion et al., 2020) and our proposed **De**tection **Co**nvNet (**DECO**). Our DECO is a simple yet effective query-based end-to-end object detection framework and enjoys the similar favorable attributes as DETR. Moreover, it is stacked with only standard convolutional layers and does not rely on any sophisticated attention modules.

Fig. 2 (b), we first upsample the object queries $o$ to obtain $\hat{o} \in \Re^{d \times H \times W}$ so that it has the same size with image feature $z_e$, *i.e.*,

$$\hat{o} = \mathrm{Upsample}(o). \tag{1}$$

There are also other design choices for the upsampling function, *i.e.*, resizing both the object queries and encoder embeddings to a fixed size before fusion, or directly upsampling the object queries to dynamic size of encoder embeddings, which is related to the resolution of input image. We will provide more analysis in experimental section. Then the upsampled object queries and the image feature embeddings are fused together using $\mathrm{Fusion}(\cdot)$ function, followed by a large kernel depthwise convolution (Liu et al., 2023b; Ding et al., 2022; 2024) to allow object queries to capture the spatial information from the image feature. An intuitive implementation for $\mathrm{Fusion}(\cdot)$ is element-wise *add* operation. There are also some alternatives like element-wise *multiply* operation, or first concatenating two features then using convolution to reduce the number of channels. More discussions will be presented in the ablation studies. The skip connection is all utilized, as shown in the following equation:

$$\hat{o}_f = \hat{o} + \mathrm{dwconv}(\mathrm{Fusion}(\hat{o}, z_e)). \tag{2}$$

The output features further go through another FFN with skip connection. Finally, an adaptive *maxpooling* is utilized to downsample the object queries back to the size of $\Re^{w_q \times h_q \times d}$ and will be further processed by the following decoder layers.

$$\hat{o}_p = \mathrm{Pooling}(\hat{o}_f + \mathrm{FFN}(\hat{o}_f)). \tag{3}$$

The final output embeddings of the decoder will be fed into the detection head to obtain the class and bounding box prediction, which is similar to the original DETR.

## 3.2 DECO: DETECTION CONVNETS EQUIPPED WITH INTERCONV

Carion *et al*. (Carion et al., 2020) proposes the Detection Transformer (DETR) that models object detection as a set prediction problem and directly produces a fixed set of objects. As shown in Fig. 3 (a), DETR first utilizes a backbone to extract image features, and feeds them into a transformer encoder and decoder architecture. A fixed small set of learned object queries interact with the global image context to directly output the final set of object predictions. DETR streamlines the end-to-end object detection pipeline and has attracted great research interest due to the good accuracy and run-time performance (Zhu et al., 2021; Zhang et al., 2022; Dai et al., 2021b;a) for object detection. Although transformers have shown great power in computer vision tasks like image classification, object detection, segmentation *etc*., there are also some recent work that reveal the potential of ConvNet-based architecture as the common backbone, *e.g*., ConvNext (Liu et al., 2022b) and ConvFormer (Yu et al., 2022). In this work we re-examine the DETR design and explore whether a ConvNet-based object detector could inherit the good properties of DETR.

One of the most important properties for DETR-based detectors is the query-based scheme for producing the final predictions. In this way, the object detector could directly obtain a fixed number of objects and gets rid of any hand-designed NMS post-processing. We follow this paradigm to design our Detection ConvNets (DECO), as shown in Fig. 3 (b). DECO also utilizes a CNN backbone

Table 1: Comparisons of our proposed DECO+ with other detectors on COCO 2017 `val` set. The FPS is measured on a V100 GPU.

| Model | Backbone | GFLOPs | FPS | AP | $AP_{50}$ | $AP_{75}$ | $AP_S$ | $AP_M$ | $AP_L$ |
|---|---|---|---|---|---|---|---|---|---|
| Faster R-CNN (Ren et al., 2015) | R50-FPN | 180 | 26 | 40.2 | 61.0 | 43.8 | 24.2 | 43.5 | 52.0 |
| Faster R-CNN (Ren et al., 2015) | R101-FPN | 246 | 20 | 42.0 | 62.5 | 45.9 | 25.2 | 45.6 | 54.6 |
| FCOS (Tian et al., 2019) | R50-FPN | 201 | 23 | 38.7 | 57.4 | 41.8 | 22.9 | 42.5 | 50.1 |
| FCOS (Tian et al., 2019) | R101-FPN | 277 | 19 | 39.1 | 58.3 | 42.1 | 22.7 | 43.3 | 50.3 |
| RetinaNet (Lin et al., 2017) | R50-FPN | 239 | 21 | 37.4 | 56.7 | 39.6 | 20.0 | 40.7 | 49.7 |
| RetinaNet (Lin et al., 2017) | R101-FPN | 315 | 17 | 38.5 | 57.6 | 41.0 | 21.7 | 42.8 | 50.4 |
| Sparse R-CNN (Sun et al., 2021b) | R50-FPN | 150 | 23 | 42.8 | 61.2 | 45.7 | 26.7 | 44.6 | 57.6 |
| OneNet-RetinaNet (Sun et al., 2021a) | R50-FPN | 239 | 21 | 37.5 | 55.4 | 40.7 | 21.5 | 40.5 | 47.4 |
| OneNet-FCOS (Sun et al., 2021a) | R50-FPN | 206 | 26 | 38.9 | 57.2 | 42.2 | 23.9 | 41.8 | 49.4 |
| DeFCN (Wang et al., 2021a) | R50-FPN | − | 19 | 41.4 | 59.5 | 45.6 | 26.1 | 44.9 | 52.0 |
| YOLOS-Ti (Fang et al., 2021) | DeiT-Tiny | 21 | 52 | 28.7 | 47.2 | 28.9 | 9.7 | 29.2 | 46.0 |
| YOLOS-S (Fang et al., 2021) | DeiT-Small | 194 | 5 | 36.1 | 55.7 | 37.6 | 15.6 | 38.3 | 55.3 |
| YOLOS-B (Fang et al., 2021) | DeiT-Base | 538 | 2 | 42.0 | 62.2 | 44.4 | 19.5 | 45.3 | 62.1 |
| DETR (Carion et al., 2020) | R50 | 97 | 28 | 39.5 | 60.3 | 41.4 | 17.5 | 43.0 | 59.1 |
| Deformable-DETR (Zhu et al., 2021) | R50 | 173 | 16 | 43.8 | 62.6 | 47.7 | 26.4 | 47.1 | 58.0 |
| Anchor-DETR (Wang et al., 2022b) | R50 | 103 | 21 | 42.1 | 63.1 | 44.9 | 22.3 | 46.2 | 60.0 |
| DAB-DETR (Liu et al., 2022a) | R50 | 94 | 25 | 42.2 | 63.1 | 44.7 | 21.5 | 45.7 | 60.3 |
| DN-DAB-DETR (Li et al., 2022) | R50 | 94 | 25 | 44.1 | 64.4 | 46.7 | 22.9 | 48.0 | 63.4 |
| Cond-DETR (Meng et al., 2021) | R50 | 90 | 28 | 43.0 | 64.0 | 45.7 | 22.7 | 46.7 | 61.5 |
| ViDT (Song et al., 2021) | Swin-Nano | 35 | 31 | 40.4 | 59.6 | 43.3 | 23.2 | 42.5 | 55.8 |
| DINO-4scale (Zhang et al., 2022) | R50 | 279 | 12 | 50.9 | 69.0 | 55.3 | 34.6 | 54.1 | 64.6 |
| Stable-DINO (Liu et al., 2023a) | R50 | − | 12 | 51.5 | 68.5 | 56.3 | 35.2 | 54.7 | 66.5 |
| SpeedDETR (Dong et al., 2023) | R50 | − | 21 | 46.8 | 66.2 | 50.4 | 28.5 | 50.6 | 63.2 |
| DDQ-DETR (Zhang et al., 2023) | R50 | − | 9 | 52.8 | 69.9 | 58.1 | 37.4 | 55.7 | 66.0 |
| Lite-DINO (Li et al., 2023) | R50 | 151 | 17 | 50.2 | - | 54.6 | 33.5 | 53.6 | 65.5 |
| Co-DETR (Zong et al., 2022) | R50 | - | 12 | 49.5 | 67.6 | 54.3 | 32.4 | 52.7 | 63.7 |
| H-DETR (Jia et al., 2022) | R50 | 280 | 19 | 50.0 | 68.3 | 54.4 | 32.9 | 52.7 | 65.3 |
| Focus-DETR (Zheng et al., 2023) | R50 | 154 | 20 | 50.4 | 68.5 | 55.0 | 34.0 | 53.5 | 64.4 |
| Rank-DETR (Pu et al., 2023) | R50 | 280 | 19 | 51.2 | 68.9 | 56.2 | 34.5 | 54.9 | 64.9 |
| RT-DETR[†] (Lv et al., 2023) | R18 | 40 | 60 | 41.7 | 61.3 | 45.3 | 25.0 | 44.0 | 56.8 |
| **DECO+ (Ours)** | R18 | 32 | **66** | 40.5 | 58.7 | 44.0 | 23.3 | 43.8 | 55.7 |
| **DECO+ (Ours)** | R50 | 69 | 34 | 47.8 | 67.1 | 52.4 | 30.6 | 51.9 | 64.2 |
| **DECO+ (Ours)** | ConvNeXt-B | 159 | 20 | **50.6** | 70.7 | 55.0 | 34.4 | 55.3 | 67.8 |

to extract features from the input image. Specifically, given a RGB image $x_{\text{img}} \in \Re^{3 \times H_0 \times W_0}$, the backbone generates feature map $f \in \Re^{C \times H \times W}$ and usually $H, W = \frac{H_0}{32}, \frac{W_0}{32}$. The feature map $f$ are then go through a CNN encoder, i.e. DECO encoder, to obtain the output embeddings $f_{enc} \in \Re^{C_e \times H \times W}$. The CNN decoder takes $f_{enc}$ as well as a fixed number of learned object queries $o \in \Re^{N \times d}$ as input to make final detection prediction via a feed forward network (FFN), where $d$ is the size of encoder output embeddings. The detailed architecture of the CNN decoder is elaborated in Sec. 3.1. We utilize the same prediction loss as in DETR, which uses bipartite matching to find paired predicted and ground truth objects.

**DECO Encoder**. Similar to DETR, a $1 \times 1$ convolution is first utilized to reduce the channel dimension of $f$ from $C$ to $d$ and obtain a new feature map $z_0 \in \Re^{d \times H \times W}$. In DETR, $z_0$ is fed into stacked transformer encoder layers, which mainly consists of multi-head self-attention (MHSA) and feed-forward network (FFN) to perform spatial and channel information mixing respectively. Recent work such as ConvNeXt (Liu et al., 2022b) has demonstrated that using stacked depthwise and pointwise convolutions could achieve comparable performance with Transformers. Therefore, we use the ConvNeXt blocks to build our DECO encoder. Specifically, each DECO encoder layer is stacked with a $7 \times 7$ depthwise convolution, a LayerNorm layer, a $1 \times 1$ convolution, a GELU acitvation and another $1 \times 1$ convolution. Besides, in DETR, positional encodings are necessary to be added to the input of each transformer encoder layer, since the transformer architecture is permutation-invariant. However, the ConvNet architecture is permutation-variant so that our DECO encoder layers could get rid of any positional encodings.

**DECO+ Equiped with Multi-scale Feature**. One limitation of original DETR as well as our DECO is the lack of multi-scale feature, which is demonstrated to be important for accurate object detection.

Table 2: Comparisons of DECO with DETR on COCO 2017. The FPS is measured on a V100 GPU.

| Model | Backbone | GFLOPs | FPS | AP | $AP_{50}$ | $AP_{75}$ | $AP_S$ | $AP_M$ | $AP_L$ |
|---|---|---|---|---|---|---|---|---|---|
| DETR (Carion et al., 2020) | R34 | 88 | 34 | 31.6 | 47.6 | 33.3 | 13.3 | 34.1 | 49.1 |
| DETR (Carion et al., 2020) | R50 | 97 | 28 | 39.5 | 60.3 | 41.4 | 17.5 | 43.0 | 59.1 |
| DETR (Carion et al., 2020) | ConvNeXt-T | 104 | 25 | 42.1 | 63.6 | 44.3 | 18.8 | 45.5 | 62.8 |
| **DECO (Ours)** | R50 | 103 | 35 | 38.6 | 58.8 | 41.1 | 19.5 | 43.3 | 55.0 |
| **DECO (Ours)** | ConvNeXt-T | 110 | 28 | 40.8 | 61.5 | 43.5 | 20.5 | 45.7 | 58.4 |

Deformable DETR (Zhu et al., 2021) utilizes the multi-scale deformable attention module to aggregate multi-scale features and obtains an improved performance compared with DETR. To equip DECO with multi-scale feature capability, we utilize the cross-scale feature-fusion module in RT-DETR (Lv et al., 2023) after obtaining the global feature from our DECO encoder. Different from RT-DETR, the features are then mapped to the same scale and concatenated along the channels followed by a linear projection layer to get the final encoder embedding. We also explore whether the performance can be further improved by utilizing deformable convolution, which is detailed in the Appendix. More modern techniques for DETRs would also be compatible with DECO, which we leave for future exploration.

## 3.3 DECO-TINYSAM: INTERCONV FOR SEGMENT ANYTHING MODEL

The mechanism of InterConv could have great potential for applying to other architectures and tasks, especially for those involving interaction across domains. As a proof of concept, we replace the mask decoder in Segment Anything Model (SAM) with our DECO decoder. The motivation is that as a prompt-based segmentation model, SAM interacts between prompt tokens and image embeddings through the mask decoder which consists of self- and cross-attentions, sharing similar spirits of our proposed SIM and CIM. More specifically, we repeat the one-dimensional queries into two dimensional features, instead of reshaping queries as in object detection task discussed above. It is mainly due to the fact that the query tokens in SAM contains learnable output tokens and a few prompt tokens, which are not reasonable to simply be reshaped into two dimensions. We utilize TinySAM (Shu et al., 2023) as the baseline model and replace the decoder to our proposed DECO architecture.

## 4 EXPERIMENTS

In this section, we first evaluate our proposed model on object detection benchmark and compare it against state-of-the-art methods. Extensive ablation studies are also conducted to provide more discussions and insights about the design choices.

### 4.1 EXPERIMENTAL SETTING

For the vanilla DECO, we follow similar training settings as DETR (Carion et al., 2020). We train the proposed DECO models for 150 epochs using AdamW optimizer, with weight decay of $10^{-4}$ and initial learning rates as $10^{-4}$ and $10^{-5}$ for the encoder-decoder and backbone, respectively. The learning rate is dropped by a factor of 10 after 100 epochs. The augmentation scheme is the same as DETR, which includes random horizontal flipping, random crop augmentation, and scale augmentation. The input image shorter side is resized to a random size between 480 and 800 pixels in the scale augmentation while restricting the longer size to at most 1333. As to DECO+ that equipped with multi-scale feature fusion, the training image size is selected between 480 and 800 with 32 stride following the RT-DETR baseline. The inference size is set to $640 \times 640$.

### 4.2 COMPARISONS WITH STATE-OF-THE-ARTS

We evaluate the proposed DECO and DECO+ on COCO benchmark and compare with recent competitive object detectors, including DETR (Carion et al., 2020), YOLOS (Fang et al., 2021), FCOS (Tian et al., 2019), and DETR variants with strong performance, *e.g.*, Anchor-DETR (Sun et al., 2021b), Conditional-DETR (Sun et al., 2021a) and ViDT. (Wang et al., 2021a) *etc.*. Experimental results in terms of detection AP and FLOPs/FPS are shown in Table 1. The FPS we report is the

| Fusion Method | GFLOPS | AP |
|---|---|---|
| Element-wise Mult. | 103 | 37.8 |
| Concat-Conv | 106 | **38.6** |
| **Element-wise Add** | **103** | **38.6** |

Table 3: Effect of different fusion methods.

| #layers | GFLOPS | FPS | AP |
|---|---|---|---|
| 5 | 101 | 35 | 38.3 |
| 6 | 103 | 35 | **38.6** |
| 7 | 105 | 34 | 38.9 |

Table 4: Effect of number of layers in decoder.

Table 5: Ablation studies for different kernel sizes in decoder.

| kernel size | 5×5 | 7×7 | **9×9** | 11×11 | 13×13 | 15×15 |
|---|---|---|---|---|---|---|
| AP (%) | 37.8 | 37.9 | **38.6** | 38.6 | 38.4 | 38.6 |
| GFLOPs | 103.35 | 103.47 | 103.58 | 103.70 | 104.11 | 104.27 |

average number of the first 100 images in the COCO 2017 *val* set on a NVIDIA V100 GPU. The FLOPs are computed with the input size of $(640, 640)$ for RT-DETR and the proposed DECO+, while $(1280, 800)$ for others. A more intuitive comparison of the trade-off between AP and latency is also shown in Fig. 1.

**Comparisons with DETR variants.** We consider different ConvNet-based backbones for DECO+ in the benchmarking. As shown in Table 1, our DECO+ with ResNet-50 Backbone achieves $47.8\%$ AP with 34 FPS on V100 GPU, which is better than most previous DETR variants considering the accuracy-latency trade-off. The ConvNeXt (Liu et al., 2022b) based DECO+ achieves an even higher AP at $50.6\%$ which is $0.2\%$ higher than Focus-DETR (Zheng et al., 2023) at the same FPS. Moreover, ResNet-18 based DECO+ obtains $40.5\%$ AP with 66 FPS, achieving quite similar performance with a variant of RT-DETR (Lv et al., 2023) that we modified to not use deformable attention and denoising training for fair comparison. Note that deformable attention and denoising training is specifically designed for attention-based architecture, and similar improved strategies for DECO still remains for future exploration.

**Comparisons with Other End-to-End Detectors.** YOLOS is an encoder-only Transformer architecture for object detection based on the vanilla pre-trained vision transformers. Our DECO+ models show clear advantage over YOLOS and have similar detection performance while running much faster. We also compare our DECO+ with recent end-to-end detectors with ConvNets, *e.g.*, Sparse R-CNN (Sun et al., 2021b), OneNet (Sun et al., 2021a) and DeFCN. (Wang et al., 2021a). As shown in Table 1, our DECO+ outperforms Sparse R-CNN (Sun et al., 2021b) and OneNet-RetinaNet (Sun et al., 2021a) with better accuracy and running speed. Similarly, DECO+ obtains $47.8\%$ AP and 34 FPS while DeFCN (Wang et al., 2021a) only has 19 FPS with $41.4\%$ AP.

**Comparisons with DETR.** We compare the performance of our vanilla DECO and DETR (Carion et al., 2020) equipped with ResNet and ConvNeXt-Tiny in Table 2. The proposed DECO encoder/decoder are adopted to substitute the transformer encoder/decoder in DETR for comparison. To align the FLOPs with DETR, we modify the DECO encoder to be three stages with the number of blocks of $(2, 6, 2)$ and the channel dimension of $(120, 240, 480)$, respectively. As shown in Table 2, for both ResNet-50 and ConvNeXt-Tiny (Liu et al., 2022b) backbones, despite higher FLOPs, our DECO obtain faster inference speed (FPS) than DETR. It demonstrates that our pure ConvNet-based architecture is more deployment-friendly than the transformer-based DETR in GPU platform. Specifically, our DECO obtains $38.6\%$ AP at 35 FPS and is $7.0\%$ AP better than DETR with ResNet-34 backbone for similar running speed.

## 4.3 ABLATION STUDIES

We conduct ablation studies based on the R50-based DECO in Table 2 to provide more discussions and insights about different design choices and justify the effectiveness of our proposed method.

**Upsampling Size in CIM.** In CIM, the object queries are first upsampled and then fused with the encoder embeddings to deal with different dimensions. Here we have different design choices, *i.e.*, resizing both the object queries and encoder embeddings to a fixed size before fusion, or directly upsampling the object queries to dynamic size of encoder embeddings, which is related to the resolution of input image. As shown in Table 6, utilizing dynamic size achieves the best performance, since it is more flexible for different input resolution and has no information discarding. Noted that $(25 \times 38)$ is the average size of COCO training set and it leads to $0.5$ AP drop than dynamic way.

| Size | GFLOPs | AP |
|---|---|---|
| $(20 \times 20)$ | 97 | 34.8 |
| $(25 \times 38)$ | 103 | 38.1 |
| $(40 \times 40)$ | 110 | 37.2 |
| Dynamic | 103 | **38.6** |

Table 6: Effect of different upsampling size of object queries in CIM.

| #queries | Query Shape ($w_q \times h_q$) | Ratio | AP |
|---|---|---|---|
| 100 | **$10 \times 10$** | 1:1 | **38.6** |
| 100 | $20 \times 5$ | 4:1 | 38.3 |
| 100 | $5 \times 20$ | 1:4 | 36.9 |
| 300 | **$30 \times 10$** | 3:1 | **38.9** |
| 300 | $20 \times 15$ | 4:3 | 38.8 |

Table 7: Effect of different shape of queries.

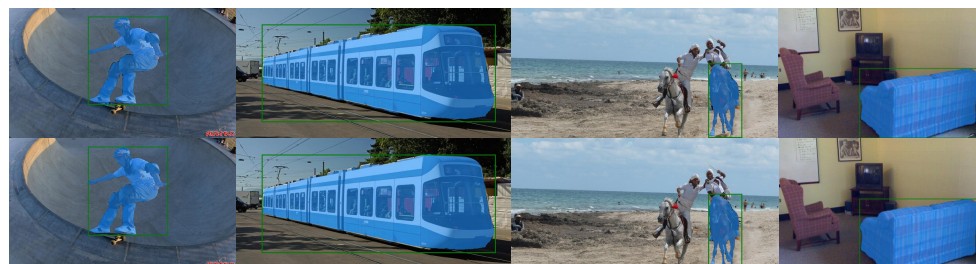

Figure 4: Visualizations for box prompted segment anything for Our DECO-TinySAM ($1^{st}$ row) and TinySAM ($2^{nd}$ row). Our method obtains quite similar performance with TinySAM.

**Number of Decoder Layers.** As shown in Table 4, more number of layers in decoder tends to have better performance. However, utilizing 6 decoder layers is a good choice to balance performance and computational cost and we keep this choice following DETR (Carion et al., 2020).

**Kernel size in decoder.** The motivation for using $9 \times 9$ dwconv is to enable sufficient receptive field. We conduct ablation experiments to explore the effect of different kernel sizes. As shown in Table 5, using $5 \times 5$ has unsatisfied performance due to limited receptive field, and enlarging kernel size to $11 \times 11$ or even $15 \times 15$ brings negligible improvement.

**Different design choices of fusion method in CIM.** As discussed in Section 3, the upsampled object queries and the image feature embeddings are fused together using *add* operations. Here we conduct ablation studies for other design choices of the fusion method, *e.g.*, using concatenation and convolution, or simple conducting element-wise multiplication for fusion. As shown in Table 3, utilizing element-wise multiplication to fuse the object queries and the image feature embeddings does not obtain better performance. Moreover, using *add* operations achieves similar detection performance with using concatenation and convolution, but has slightly smaller FLOPs.

**Different shapes of object queries.** In our proposed method, the object queries should be in 2D shape of $w_q \times h_q$ and there are several choices of query shape for $N$ object queries. For example, $w_q \times h_q$ could be $10 \times 10$, $20 \times 5$ or $5 \times 20$ for $N = 100$ queries. As shown in Table 7, using $10 \times 10$ obtains better detection performance. When $N = 300$, using query shape of $30 \times 10$ achieves slightly better performance than $20 \times 15$. A typical ratio of image size for COCO could be considered as $1333 : 800 \approx 1.67$ and we could conclude from Table 7 that better performance is obtained when the query shape is approximately the ratio of input image.

## 4.4 EXTENSION TO SEGMENT ANYTHING TASK

Our proposed DECO is original designed for object detection. However, the mechanism of DECO decoder could have great potential for applying to other architectures and tasks, especially for those involving interaction across domains. As a proof of concept, we replace the mask decoder in

| Method | COCO AP (%) | Mobile Lat. (ms) |
|---|---|---|
| TinySAM (Shu et al., 2023) | 41.9 | 34 |
| DECO-TinySAM | 41.4 | 29 |

Table 8: Zero-shot instance segmentation results for TinySAM baseline and our DECO-TinySAM.

Segment Anything Model (SAM) with our DECO decoder. The motivation is that as a prompt-based segmentation model, SAM interacts between prompt tokens and image embeddings through the mask decoder which consists of self- and cross-attentions, sharing similar spirits of our proposed self-InterConv and cross-InterConv. More specifically, we repeat the one-dimensional queries into two dimensional features, instead of reshaping queries as in object detection task discussed above. It

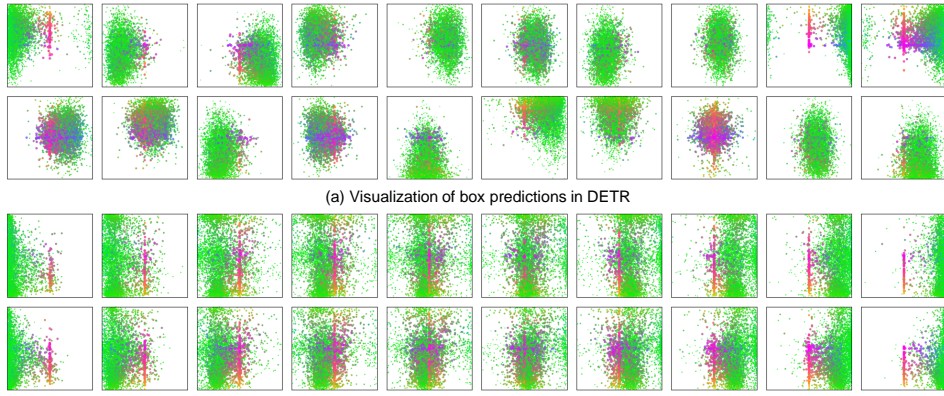

(a) Visualization of box predictions in DETR

(b) Visualization of all box predictions in DECO

Figure 5: Visualizations of query slots for DETR and our DECO. We can find that both DECO and DETR tend to make different object queries to focus on different patterns in terms of spatial areas and box sizes. Interestingly, we could also observe that the slots of DETR for small objects are spatially unordered while the distributions of each slot for our DECO are spatially ordered for small boxes.

is mainly due to the fact that the query tokens in SAM contains learnable output tokens and a few prompt tokens, which are not reasonable to be simply reshaped into two dimensions.

We utilize TinySAM (Shu et al., 2023) as the baseline model and replace the decoder to our proposed InterConv architecture. The results for zero-shot instance segmentation prompted by detected boxes on COCO dataset are shown in Table 8. Despite without carefully tuning, our DECO-TinySAM obtains quite similar zero-shot instance segmentation performance with that of TinySAM baseline, with lower latency on mobile devices. We also provide some visualizations for box prompted segment anything for our DECO-TinySAM and the TinySAM baseline, demonstrating strong performance and generalizability to other tasks of our method.

## 4.5 VISUALIZATION

**Visualization of Query Slots.** Following the same method in DETR, we visualize the boxes predicted by 20 out of total 100 query slots of our DECO. Each point represents one bounding box prediction and the coordinates are normalized by each image size. Different colors indicate objects with different scales, *e.g.*, green, red and blue refer to small boxes, large horizontal boxes and large vertical boxes, respectively. As shown in Fig. 5, we can find that both DECO and DETR tend to make different object queries to focus on different patterns in terms of spatial areas and box sizes. Interestingly, we could also observe that the slots of DETR for small objects are spatially unordered, which indicates that the prediction of each slot is random in spatial dimension. However, things are a bit different for our DECO, whose distributions of each slot are spatially ordered for small boxes. This observation is most likely to be related to the cross-interaction mechanism of object queries and image features, where cross-attention module tends to capture global information and our proposed module tends to focus on local interaction through large kernel convolutions.

## 5 CONCLUSION AND DISCUSSION

In this paper, we aim to explore whether we could build query-based detection and segmentation framework with ConvNets instead of sophisticated transformer architecture. We propose a novel mechanism dubbed InterConv to perform interaction between object queries and image features via convolutional layers. Equipped with the proposed InterConv, we build Detection ConvNet (DECO), which is composed of a backbone and convolutional encoder-decoder architecture. We compare the proposed DECO against prior detectors on the challenging COCO benchmark. Despite its simplicity, our DECO achieves competitive performance in terms of detection accuracy and running speed. Specifically, with the ResNet-18 and ResNet-50 backbone, our DECO achieves $40.5\%$ and $47.8\%$ AP with 66 and 34 FPS, respectively. The proposed method is also evaluated on the segment anything task, demonstrating similar performance and higher efficiency. We hope the proposed method brings another perspective for designing architectures for vision tasks.

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
