# DECO: Unleashing the Potential of ConvNets for Query-based Detection and Segmentation –Supplementary material–

**Xinghao Chen**[1]*, **Siwei Li**[1,2]*, **Yijing Yang**[1], **Yunhe Wang**[1]

[1] Huawei Noah's Ark Lab   [2] Tsinghua University
{xinghao.chen, yunhe.wang}@huawei.com

## A   Appendix

### A.1   Improve DECO with Deformable Convolution

As discussed in the paper, some advanced DETR variants adopt deformable attention to aggregate the multi-scale features. As for the proposed convolutional framework, we explore whether deformable convolution can further boost the performance of DECO+. We replace the dwconv in Self-InterConv with a block of DCNv3. As shown in Table 1, utilizing deformable convolution brings +1.0 AP gain and even obtains slightly faster FPS, from 66 to 67.

### A.2   Compare DECO with DETR equipped with FlashAttention

There are also some popular attempts for speeding up attention module like FlashAttention. The FPS for DETR reported in the main manuscript is measured in Pytorch 1.10 without FlashAttention. We utilize XFormers to adopt flash attention to DETR and measure the latency on V100 GPU. It should be noted that flash attention is only supported in Pytorch 2.0+ and our vanilla DECO also obtains higher FPS in this setting. As shown in Figure 1, although DETR obtains more speedup by using Pytorch 2.1 and FlashAttention, our DECO still achieves better trade-off with respect to FPS and AP.

Table 1: Comparison between DECO+ with and without deformable convolution in Self-InterConv.

| Model | Backbone | FPS | AP) |
|---|---|---|---|
| DECO+ | R18 | 66 | 40.5 |
| DECO+ w/ DCNv3 | R18 | 67 | 41.5 |

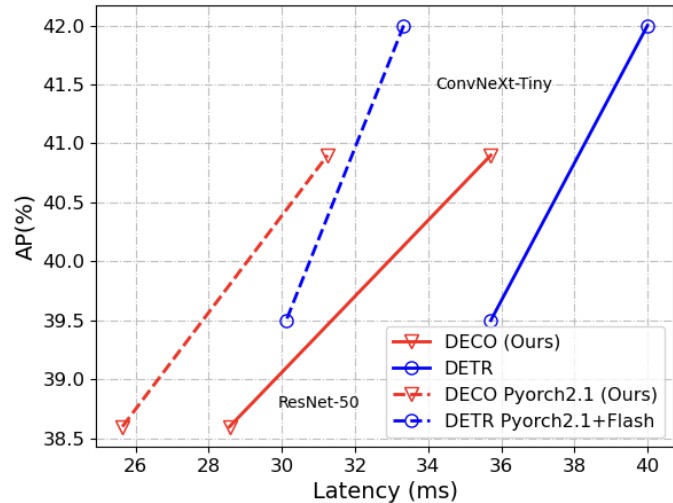

Figure 1: Comparisons between vanilla DECO and DETR+FlashAttention.

---

\* Equal contribution. Correspondence to X. Chen and Y. Wang.