# OpenReview forum: "DECO: Unleashing the Potential of ConvNets for Query-based Detection and Segmentation"
_ICLR.cc/2025/Conference — ICLR 2025 Poster_

### Official Review · Reviewer_3qSX · 2024-11-02

**Soundness:** 3
**Presentation:** 3
**Contribution:** 3
**Rating:** 6
**Confidence:** 4

**Summary:**

The authors introduced a CNN-based end-to-end detection architecture, DECO. DECO contains an encoder and a decoder with pure convolutions nad is easy to implement. The experimental results demonstrate the effectiveness.

**Strengths:**

+ The paper is well-written and easy to follow.
+ Experiments are strong, demonstrating the effectivenss of CNN-based model.

**Weaknesses:**

+  My main concern is the use of cross-scale feature-fusion module which is proposed in RT-DETR, which weakens the contribution. It would make the paper more strong if the authors could further discuss the difference.
+ There are many recent studies speeding up transformers, it would be nice to discuss more on using convolutions. Claims in L073-L074 are not convincing to me.

**Questions:**

NA

---

> ### Author Response · Authors · 2024-11-22
> **Response to Reviewer 3qSX**
>
> _**Q1: My main concern is the use of cross-scale feature-fusion module which is proposed in RT-DETR, which weakens the contribution. It would make the paper more strong if the authors could further discuss the difference.**_
>
> **A1**: Thanks for the concern. We would like to highlight that our DECO also outperform DETR with the same settings, *i.e.*, training receipt, architecture etc. The comparisons are shown in Table 2 (as also shown in Figure 1 in supplementary material) and we can see that our DECO obtains better performance than DETR, which justifies the effectiveness and also the main contribution of our proposed method.
>
> Due to the powerful capacity of DETR framework, lots of follow-up methods have improved DETR from different perspectives to greatly boost the performance and RT-DETR is one of the excellent work. To make more proper comparisons with state-of-the-art methods, we incorporate the multi-scale structure of RT-DETR into our DECO framework and obtains stronger performance than most DETR variants. The core DECO encoder module for extracting global feature is different from RT-DETR. Besides, as stated in the paper Line 335-338, the combination of features between different scales is also different from RT-DETR.
>
> We would like to argue that it actually showcases our DECO is a general framework to be compatible with other modules and more explorations are left for future work.
>
> _**Q2: There are many recent studies speeding up transformers, it would be nice to discuss more on using convolutions. Claims in L073-L074 are not convincing to me.**_
>
> **A2**: Thanks for the question.
> - There are quite a few studies for speeding up detection transformer (DETR) like Focus-DETR, Lite-DETR etc. As shown in Figure 1, our proposed method obtains similar accuracy/latency tradeoff with Focus-DETR and outperforms Lite-DETR, demonstrating the strong performance of our model.
> - There are also some popular attempts for speeding up attention module like FlashAttention. The FPS for DETR reported in the manuscript is measured in Pytorch 1.10 without FlashAttention. We utilize XFormers to adopt flash attention to DETR and measure the latency on V100 GPU. It should be noted that flash attention is only supported in Pytorch 2.0+ and our DECO also obtains higher FPS in this setting. Though DETR obtains more speedup by using Pytorch 2.1 and FlashAttention, our DECO still achieves better tradeoff with respect to  FPS and AP, as shown in Figure 2 in supplementary material.

---

> ### Comment · Reviewer_3qSX · 2024-11-26
>
> Thanks for the rebuttal. My concerns are well addressed and I'm increasing my score.

---

> > ### Author Response · Authors · 2024-11-26
> >
> > We sincerely thank you for your thoughtful feedback and positive evaluation. Your comments have been crucial in improving this manuscript, and we greatly appreciate the insights you have shared. Thank you again for your support!

---

### Official Review · Reviewer_wv8n · 2024-11-04

**Soundness:** 3
**Presentation:** 3
**Contribution:** 3
**Rating:** 5
**Confidence:** 5

**Summary:**

The paper presents a new approach to object detection and segmentation using Convolutional Neural Networks (ConvNets), challenging the dominance of transformer-based architectures in these fields. This paper introduces InterConv, a mechanism that mimics the interaction between object queries and image features, traditionally handled by attention mechanisms in transformers, but using only convolutional layers. This paper presents Detection ConvNet (DECO), an end-to-end object detection framework that replaces transformers with ConvNets. The proposed DECO obtains comparable results compared to transformer-based methods.

**Strengths:**

1. This paper presents InterConv which mimics attention mechanism with convolutional layers, and formulates Self-Interaction Module and Cross-Interaction Module (CIM) as self-attention and cross-attention.
2. This paper explores a convolution-based DETR framework with the proposed InterConv.
3. The proposed DECO obtains comparable performance compared to DETR.

**Weaknesses:**

1. Although this paper presents the convolution-version DETR which replaces attentions with convolutions, the overall performance is inferior to recent transformer-based methods, such as DINO, Co-DETR, and Stable-DINO[1], showing the limitations of using convolutional architectures.
2. In Tab. 2, it shows that DETR achieves better performance than DECO on both R50 and ConvNeXt-T while the improvements in inference speeds are not significant. In addition, DETR can be further optimized for acceleration therefore obtaining better inference speed and accuracy. Hence, the effectiveness of the proposed method in this paper is rather limited.
3. This paper lacks experimental comparisons with many recent works, such as Stable DINO[1], RT-DETRv2[2], DDQ[3], and SpeedDETR[4], in terms of both accuracy and speed.
4. It's hard to transfer the well-established techniques of recent DETR variants, such as denoising queries and deformable attention, into DECO for better performance.
5. The proposed object queries require prior knowledge to determine the best shape/layout, for example, 30x10 queries for the COCO dataset. I'm very concerned about whether the pre-trained detector DECO can perform well on the other datasets with different aspect ratios, such as datasets with 1:3 aspect ratios.
6. The details of extending DECO to segmentation tasks are unclear.



References\
[1] Liu et al. Stable-DINO: Detection Transformer with Stable Matching. ICCV 2023.\
[2] Lv et al. RT-DETRv2: Improved Baseline with Bag-of-Freebies for Real-Time Detection Transformer. arXiv 2024.\
[3] Zhang et al. Dense Distinct Query for End-to-End Object Detection. CVPR 2023.\
[4] Dong et al. SpeedDETR: Speed-aware Transformers for End-to-end Object Detection. ICML 2023.

**Questions:**

1. I'm concerned about the initialization of the object queries. Could the object queries be random initialized variables for different images?
2. In Sec. 4.4, does this paper combine DECO with TinySAM?

---

> ### Author Response · Authors · 2024-11-22
> **Response to Reviewer wv8n**
>
> _**Q1: Although this paper presents the convolution-version DETR which replaces attentions with convolutions, the overall performance is inferior to recent transformer-based methods, such as DINO, Co-DETR, and Stable-DINO[1], showing the limitations of using convolutional architectures.**_
>
> **A1**: Thanks for the comments. We have already included the comparisons with DINO and Co-DETR in our original manuscript (Figure 1 and Table 1). As suggested, we also include the results of Stable-DINO in the updated manuscript and also highlight the comparisons of these models below.
>
> | Model   | FPS  | AP        | AP_50 | AP_75 | AP_S | AP_M | AP_L |
> |------------------|------------|--------|---------|-----------|-----------|-----------|----------|
> | DINO-4scale | 12    | 50.9    | 69.0    | 55.3    | 34.6   | 54.1   | 64.6   |
> | Stable-DINO-4scale | 12    | 51.5    | 68.5    | 56.3    | 35.2   | 54.7   | 66.5   |
> | Co-DETR   | 12      | 49.5      | 67.6      | 54.3      | 32.4     | 52.7     | 63.7     |
> | DECO+ (Ours)  | **20**    | **50.6** | 70.7    | 55.0    | 34.4   | 55.3   | 67.8   |
>
> As shown in the above table, our model actually obtains better AP than Co-DETR and comparable AP with DINO and Stable-DINO, but with much faster inference speed. A more clear visualization in Figure 1 of the manuscript also demonstrates that our model achieve better accuracy and latency trade-off than these models.
>
> _**Q2: In Tab. 2, it shows that DETR achieves better performance than DECO on both R50 and ConvNeXt-T while the improvements in inference speeds are not significant. In addition, DETR can be further optimized for acceleration therefore obtaining better inference speed and accuracy. Hence, the effectiveness of the proposed method in this paper is rather limited.**_
>
> **A2**: Thanks for the concern. We would like to highlight that our DECO also outperforms DETR with the same settings, *i.e.*, training receipt, architecture etc. The comparisons are shown in Table 2 (as also shown in Figure 1 in supplementary material) and we can see that our DECO obtains better performance than DETR, which justifies the effectiveness and also the main contribution of our proposed method.
>
> There are also some popular attempts for speeding up attention module like FlashAttention. The FPS for DETR reported in the manuscript is measured in Pytorch 1.10 without FlashAttention. We utilize XFormers to adopt flash attention to DETR and measure the latency on V100 GPU. It should be noted that flash attention is only supported in Pytorch 2.0+ and our DECO also obtains higher FPS in this setting. Though DETR obtains more speedup by using Pytorch 2.1 and FlashAttention, our DECO still achieves better tradeoff with respect to  FPS and AP, as shown in Figure 2 in supplementary material.
>
>
> _**Q3: This paper lacks experimental comparisons with many recent works, such as Stable DINO[1], RT-DETRv2[2], DDQ[3], and SpeedDETR[4], in terms of both accuracy and speed.**_
>
> **A3**: As suggested, we include the results of Stable-DINO, DDQ, SpeedDETR in the updated manuscript and also highlight the comparisons of these models below.
>
> | Model   | FPS  | AP        | AP_50 | AP_75 | AP_S | AP_M | AP_L |
> |------------------|------------|--------|---------|-----------|-----------|-----------|----------|
> | Stable-DINO-4scale | 12    | 51.5    | 68.5    | 56.3    | 35.2   | 54.7   | 66.5   |
> | SpeedDETR |21 | 46.8 | 66.2 | 50.4 | 28.5 | 50.6 | 63.2|
> | DDQ-DETR | 9 | 52.8 | 69.9 | 58.1 | 37.4 | 55.7 | 66.0|
> | DECO+ (Ours)  | **20**    | **50.6** | 70.7    | 55.0    | 34.4   | 55.3   | 67.8   |
>
> As shown in the above table, our model obtains similar FPS with SpeedDETR and has 3.8% higher AP. Though our DECO has slightly lower AP than Stable-DINO and DDQ, it obtains much faster inference speed. A more clear visualization in Figure 1 of the manuscript also demonstrates that our model achieve better accuracy and latency trade-off than these models.
>
> RT-DETRv2 is a strong real-time detector which mainly introduce modifications within the deformable attention module and improved training strategy to achieve enhanced performance. According to the results of DCNv3 presented in the response of ***Q4***, DECO can gain from using deformable convolution. Thus, this bag of freebies for training strategy in RT-DETRv2 can also be generalized to our DECO framework, including those related to deformable attention. We will explore more in future work.
>
> We also would like to highlight that this paper aims to explore an alternative solution using CNNs and prove that it can also achieve quite competitive performance with Transformers, in terms of both object detection and segment anything tasks. We hope that this fresh idea of convolutional query-image fusion architecture can shed some light on the future design. Our DECO is a general framework to be compatible with other modules and more explorations are left for future work.

---

> > ### Author Response · Authors · 2024-11-22
> > **Response to Reviewer wv8n (continued)**
> >
> > _**Q4: It's hard to transfer the well-established techniques of recent DETR variants, such as denoising queries and deformable attention, into DECO for better performance.**_
> >
> > **A4**: Thanks for this interesting point.
> >
> > The denoising training in DN-DETR and DINO aims at speeding up the convergence of bipartite matching. It introduces the noised ground-truth bounding boxes with their labels into the training process to make the matching loss more stable. To avoid leakage of ground truth, attention mask is used to separate the real object queries and the denoising ones.
> > As this bipartite matching is also used in DECO, we also face the need to speed up the convergence. Denoising training is possible to be applied to our convolutional design in a similar way. For example, the denoising groups of queries can be inserted into the initial object queries. The number of denoising queries can be controlled to ensure the entire queries are able to be reshaped to a 2D arrangement. To avoid knowledge leakage, similar ideas of masked convolutions can be adopted in the training process.
> >
> > We replace the dwconv in Self-Interaction Module (SIM) with deformable convolution, using the implementation of DCNv3. As shown in the below table, utilizing deformable convolution brings **+1.0 AP** gain and even obtains slightly faster FPS (**66 vs. 67**). We are  trying to also use deformable convolution in Cross-Interaction Module (CIM) and hoping to obtain further improvement.
> >
> > | Model   | Backbone | FPS  | AP        |
> > |------------------|------------|--------|--------|
> > |DECO+| R18 | 66 | 40.5 |
> > |DECO+ **w/ DCNv3** | R18 | 67 | **41.5** |
> >
> > _**Q5: The proposed object queries require prior knowledge to determine the best shape/layout, for example, 30x10 queries for the COCO dataset. I'm very concerned about whether the pre-trained detector DECO can perform well on the other datasets with different aspect ratios, such as datasets with 1:3 aspect ratios.**_
> >
> > **A5**: Thanks for this interesting point. As shown in Table 7, we empirically find that better performance is obtained when the query shape is approximately the ratio of input image. However, we can also see that for 100 queries, even we utilize the most dissimilar ratio like 1:4, the detection performance on COCO only suffers from 1.4% drop. Therefore, our model is kind of robust to the shape of object query.
> >
> > _**Q6: The details of extending DECO to segmentation tasks are unclear. In Sec. 4.4, does this paper combine DECO with TinySAM?**_
> >
> > **A6**: Thanks for the question. Yes, we generalize the proposed InterConv in DECO to TinySAM for the prompt-based segmentation task. As our main contribution, the design of the fully convolutional SIM and CIM modules in our proposed InterConv act as the same role of self-attention and cross-attention modules in Transformers. Specifically, both SIM and CIM together perform the interaction between object queries and image features, which looks for expected objects from an image using the learnt queries. On the other hand, the lightweight mask decoder in SAM has the same spirit. It interacts between the output/prompt tokens and the image embeddings through attention blocks. This motivates us to apply our proposed methodology to this prompt-based segmentation task in the experiments, using TinySAM as the baseline.
> >
> > The visualization of the segmentation results is to show the effectiveness of this application. Our convolutional solution of the mask decoder shows competitive segmentation precision, meanwhile achieving a significant latency reduction on mobile devices. The experiment proves that our proposed DECO decoder is a general solution for the scenarios that requires query/prompt and image interaction, especially on edge devices.
> >
> > We have include all these details in the updated manuscript.
> >
> > _**Q7: I'm concerned about the initialization of the object queries. Could the object queries be random initialized variables for different images?**_
> >
> > **A7**: Thanks for raising this point. The object queries in DETR are randomly initialized and fixed after training. Our DECO shares similar initialization scheme with prior methods. It should be noted that both this initialization strategy is not relevant with input images. Several subsequent methods like RT-DETR utilize query selection schemes, which use confidence score to select the top K features from the encoder to initialize object queries.

---

> > > ### Comment · Reviewer_wv8n · 2024-11-25
> > >
> > > Thank you very much to the authors for providing a detailed response. Although the authors have presented experiments and comparisons to demonstrate the method's advantages in terms of the speed-accuracy trade-off, I still have concerns about the method's scalability (similar to DETR), the predefined shape of object queries, and the upper limits of the approach.
> > >
> > > Currently, the DETR series of methods have become relatively mature, and exploring the replacement of attention with convolution is one research direction. However, its impact on the development of object detection remains unclear, and its advantages are not particularly prominent. I would like to hear the authors' discussion regarding the use of convolution to replace attention.

---

> > > > ### Author Response · Authors · 2024-11-26
> > > >
> > > > Thanks for the feedback. We sincerely appreciate the reviewer’s acknowledgment of the advantages of our method in terms of the speed-accuracy trade-off. We believe that this balance is crucial for practical applications. We would like to further provide some discussions about the reviewer's concerns and hope to achieve more clarifications.
> > > >
> > > > 1. First we would like to emphisize that our proposed method has **good scalability, in terms of both across different backbones and different tasks**. As discussed in the manuscript, our proposed method has demonstrated effectiveness in both classical object detection and segment anything tasks. We also include more results (the last row in below table) to demonstrate the scalability of larger backbones. We can see that when the backbone scales from ConvNeXt-B to ConvNeXt-L, our model could still obtain +0.7% AP. As a reference, ConvNeXt-L backbone brings +0.8% AP for COCO than ConvNeXt-B with Cascade Mask R-CNN as reported in ConvNeXt paper. These results also somehow justify that **the upper limits of our approach is not worse than prior work.**
> > > >
> > > > | Model   | Backbone | FPS  | AP        |
> > > > |------------------|------------|--------|--------|
> > > > |DECO+| R18 | 66 | 40.5 |
> > > > |DECO+ | R50 | 34 | **41.5** |
> > > > |DECO+| ConvNeXt-B | 20 | 50.6 |
> > > > |DECO+ | ConvNeXt-L | 13 | **51.3** |
> > > >
> > > > 2. Meanwhile, we also would like to point out that **the predefined shape of object query does not impact its practical applications**.  In most cases we could obtain the shape of input image since it is actually a pre-defined value, e.g., 2048x1024 for Cityscapes, 1024x800 for OpenImages, and 1333x800 for Objects365 according to the typical settings of MMDetection. Training on a specific dataset and directly evaluating on other domain of data with quite different ratio is a common challenge in object detection. Our model has demonstrated some kind of robustness to the shape of object query.
> > > >
> > > > 3. Finally, we would like to provide **more discussions and motivations for our work**. As pointed out by the reviewer, the community of computer vision recently pays more attention for bringing back convolutions instead of transformers, *i.e.*, ConvNeXt, MetaFormer etc.
> > > > As convolutional operations are more deployment-friendly in most hardware platforms, we attempt to build a query-based encoder-decoder style detector with pure convolution, by carefully design a cross interaction module consists of convolution layers. Experiments in the paper show the effectiveness and efficiency of DECO/DECO+ compared to other transformer-based methods in terms of accuracy, computational cost and latency on both GPU and mobile chips. It is proved to be more efficient than transformer-based solutions in both detection tasks and the application to prompt-based segmentation problems.
> > > > In conclusion, we try to propose an alternative solution using CNNs and prove that it can also achieve quite competitive performance with Transformers. We hope that this fresh idea of convolutional query-image fusion architecture can shed some light on the future design.

---

> > > > > ### Author Response · Authors · 2024-11-27
> > > > > **Additional experiments for predefined object query shape**
> > > > >
> > > > > As an additional experiment for the concern of predefined object query shape, we evaluate DECO trained under different query shapes on a subset of COCO2017 validation set, as shown in the table below. The subset contains 537 vertical images with $width:height< 1:1.45$. We choose DETR and OneNet-FCOS as comparison since they achieve similar performance on COCO. It shows that DECO trained with 1:1 and 3:1 (square or horizontal) object query aspect ratios perform similarly on vertical images, and can outperform or being comparable with OneNet-FCOS. It indicates that a predefined object query shape can perform well on dataset with a different aspect ratio and does not impact its practical applications.
> > > > >
> > > > > | Model       | Backbone | Query Shape $w_q\times h_q$ | AP (COCO2017 val) | AP   (subset with $width:height< 1:1.45$) |
> > > > > | ----------- | -------- | --------------------------- | ----------------- | ------------- |
> > > > > | DETR        | R50      | -                           | 39.5              | 46.1          |
> > > > > | OneNet-FCOS | R50      | -                           | 38.9              | 43.6          |
> > > > > | DECO        | R50      | 10x10 (1:1)                 | 38.6              | 44.9          |
> > > > > | DECO        | R50      | 30x10 (3:1)                 | 38.9              | 44.8          |
> > > > >
> > > > > We hope that the experiments and discussions could address the reviewer’s concerns. We would also appreciate any further feedback to continue improving our work.

---

### Official Review · Reviewer_yhFU · 2024-11-06

**Soundness:** 4
**Presentation:** 4
**Contribution:** 3
**Rating:** 8
**Confidence:** 5

**Summary:**

This paper proposes a query-based end-to-end Encoder-Decoder architecture, i.e., DECO with pure CNN modules for object detection, built upon a mechanism dubbed InterConv for the interaction between object queries and image features. Experimental results show that the proposed method without complicated components achieves competitive performance and efficiency over the DETR series.

**Strengths:**

1. The proposed InterConv consisting of pure convolutional layers can help the encoder and decoder capture long-range details without attention mechanisms and also help the decoder interact with object queries.
2. Without many complicated designs, DECO achieves comparative performance and efficiency over DETR.
3. DECO can be further improved to DECO+ by introducing the multi-scale design from Deformable-DETR.

**Weaknesses:**

1. On Lines 074-075, within the scope of object detection, the statement ``whose weights and inputs are both generated during runtime'' is unclear, as the only ones dynamically generated are intermediate variables (including attention weights).
2. The performance can be further improved using more convolutional-based techniques like deformable convolutions.

**Questions:**

1. Can the denoising training technique in DN-DETR and DINO be introduced into convolutional-based DECO?
2. To further improve the performance, it may be practical to introduce the deformable convolution (v1-v4) into the encoder and decoder. What about the efficiency?

---

> ### Author Response · Authors · 2024-11-22
> **Response to Reviewer yhFU**
>
> _**Q1: On Lines 074-075, within the scope of object detection, the statement ``whose weights and inputs are both generated during runtime'' is unclear, as the only ones dynamically generated are intermediate variables (including attention weights).**_
>
> **A1**: Thanks for the question. To be more clear, we aim to emphasize that the **attention mechanism introduces dynamic memory access where both attention weights and inputs are generated during runtime**. In contrast, **the weights in convolution is static and only the inputs are dynamic**. The above discussions are related to the attention and convolution operations, and are actually decoupled with the tasks. Therefore, this statement is equally valid within the scope of object detection.
>
> _**Q2: The performance can be further improved using more convolutional-based techniques like deformable convolutions.**_
>
> **A2**: Thanks for the good suggestion. We replace the dwconv in Self-Interaction Module (SIM) with deformable convolution, using the implementation of DCNv3. As shown in the below table, utilizing deformable convolution brings **+1.0 AP** gain and even obtains slightly faster FPS (**66 vs. 67**). We are  trying to also use deformable convolution in Cross-Interaction Module (CIM) and hoping to obtain further improvement.
>
> | Model   | Backbone | FPS  | AP        |
> |------------------|------------|--------|--------|
> |DECO+| R18 | 66 | 40.5 |
> |DECO+ **w/ DCNv3** | R18 | 67 | **41.5** |
>
> _**Q3: Can the denoising training technique in DN-DETR and DINO be introduced into convolutional-based DECO?**_
>
> **A3**: Thanks for the question. The denoising training in DN-DETR and DINO aims at speeding up the convergence of bipartite matching. It introduces the noised ground-truth bounding boxes with their labels into the training process to make the matching loss more stable. To avoid leakage of ground truth, attention mask is used to separate the real object queries and the denoising ones.
>
> As this bipartite matching is also used in DECO, we also face the need to speed up the convergence. Denoising training is possible to be applied to our convolutional design in a similar way. For example, the denoising groups of queries can be inserted into the initial object queries. The number of denoising queries can be controlled to ensure the entire queries are able to be reshaped to a 2D arrangement. To avoid knowledge leakage, similar ideas of masked convolutions can be adopted in the training process.

---

> > ### Comment · Reviewer_yhFU · 2024-11-27
> > **Official Comment by Reviewer yhFU**
> >
> > Thanks for the authors' response; it addresses all of my concerns. I decide to keep the positive rating. The updated results and explanations can be further added to the paper. This work is worth further exploring in the future.

---

> > > ### Author Response · Authors · 2024-11-27
> > >
> > > Thank you for your positive feedback and  rating. We are glad that our response has addressed your concerns. We will carefully incorporate the updated results and explanations into the paper.

---

### Official Review · Reviewer_8onY · 2024-11-10

**Soundness:** 4
**Presentation:** 3
**Contribution:** 3
**Rating:** 8
**Confidence:** 4

**Summary:**

The paper proposes an InterConv mechanism to facilitate interaction between object queries and image features via a convolutional layer, resulting in the Detection ConvNet (DECO). DECO is composed of a backbone and a convolutional encoder-decoder architecture. Results on COCO detection demonstrate that DECO achieves a decent accuracy-speed trade-off.

**Strengths:**

- The DECO method requires only standard convolutions, making it computationally efficient and more compatible with low-end AI chips compared to DETR-based methods.

- The designs of self-interconv and cross-interconv are extremely simple and effective, indicating that the paper captures the essence of DETR and presents an elegant pure-conv replacement.

**Weaknesses:**

- Clarity: In the method section, the Fusion operation is crucial; however, the paper fails to discuss it adequately. In the experiment section, the paper presents different fusion methods and query map upsampling strategies, but they remain unclear. These important details should be clearly stated and explained in the method section.

- The paper is missing some important references, such as the large kernel CNN paper and the SparseInst paper, which is also a pure-conv query-based object detection/segmentation method.

**Questions:**

Please help to address the clarity issues.

**Details Of Ethics Concerns:**

No Ethics Concerns

---

> ### Author Response · Authors · 2024-11-22
> **Response to Reviewer 8onY**
>
> _**Q1: Clarity: Fusion methods and query map upsampling strategies should be clearly stated and explained in the method section.**_
>
> **A1**: Thanks for your suggestions. We provide more details in the method section for the fusion and upsampling strategies in Line 237-246.
> > **There are also other design choices for the upsampling function, \ie, resizing both the object queries and encoder embeddings to a fixed size before fusion, or directly upsampling the object queries to dynamic size of encoder embeddings, which is related to the resolution of input image. We will provide more analysis in experimental section.**
> Then the upsampled object queries and the image feature embeddings are fused together using **$\mathrm {Fusion}(\cdot)$ function**, followed by a large kernel depthwise convolution to allow object queries to capture the spatial information from the image feature.
> **An intuitive implementation for $\mathrm {Fusion}(\cdot)$ is element-wise *add* operation. There are also some alternatives like element-wise *multiply* operation, or first concatenating two features then using convolution to reduce the number of channels. More discussions will be presented in the ablation studies.**
>
> _**Q2: The paper is missing some important references, such as the large kernel CNN paper and the SparseInst paper, which is also a pure-conv query-based object detection/segmentation method.**_
>
> **A2**: Thanks for pointing out. We have included related papers regarding large kernel convolutions[1,2,3]. We utilize a large kernel depth-wise convolution in the proposed InterConv module, to enhance the interaction of image embeddings and query features.
>
> We also include detailed discussions with SparseInst in related work section.
> > **Cheng et. al proposed SparseInst,  an efficient and fully convolutional framework for real-time instance segmentation. SparseInst utilizes a sparse set of instance activation maps to predict objects in end-to-end style, which could be viewed as an alternative for the query-based mechanism of DETR**. In this paper we would like to design a DETR-like detection pipeline but built with standard convolutions, which could inherit both the advantages of ConvNets and the favorable properties {of the} DETR framework. **Compared with SparseInst, our method follows the similar query-base mechanism of DETR, and also demonstrates good generalization capability like segment anything models.**
>
> [1] Ding et. al, Scaling up your kernels to 31x31: Revisiting large kernel design in cnns, CVPR 2022\
> [2] Liu et. al, More convnets in the 2020s: Scaling up kernels beyond 51x51 using sparsity, ICLR 2023\
> [3] Ding et. al, UniRepLKNet: A Universal Perception Large-Kernel ConvNet for Audio Video Point Cloud Time-Series and Image Recognition, CVPR 2024\
> [4] Cheng et. al, Sparse Instance Activation for Real-Time Instance Segmentation, CVPR 2022

---

> > ### Comment · Reviewer_8onY · 2024-12-03
> >
> > Thanks for the authors' response. I keep my original rating. Regarding your A1, I would like to suggest you update your paper and post a new version.

---

> > > ### Author Response · Authors · 2024-12-04
> > >
> > > Thank you for your positive feedback and rating. We have included all the points raised in A1 in the revised version. Additionally, we will continue to refine and improve the manuscript further.

---

### Meta-Review · Area_Chair_d1oL · 2024-12-18

**Metareview:**

After discussion, this submission received 3 positive scores and a negative score. The reviewer who assigned the negative score raised unresolved issues about efficiency, accuracy and scalability of the proposed detector. After reading the paper, the review comments and the rebuttal, the AC thinks that this study has significance to the object detection community: utilizing pure convolution to build a transformer-style decoder module could inspire future researches. The AC supports the acceptance of this submission as a poster paper in ICLR.

**Additional Comments On Reviewer Discussion:**

Three of the reviewers thought that their concerns were solved while The reviewer who assigned the negative score raised unresolved issues about detector efficiency, accuracy and scalability of the proposed detector. The AC agrees with reviewer #wv8n upon these issues but thinks that a good research is about the future, $i.e.$, inspiring new researches along this direction. Therefore, the AC recommends to accept this submission given its average score higher than the threshold bar.

---

### Decision · Program_Chairs · 2025-01-22

Accept (Poster)